# Effects of the Species and Growth Stage on the Antioxidant and Antifungal Capacities, Polyphenol Contents, and Volatile Profiles of Bamboo Leaves

**DOI:** 10.3390/foods13030480

**Published:** 2024-02-02

**Authors:** Hui Shen, Yan Wang, Pingping Shi, Hong Li, Yanan Chen, Tenggen Hu, Yuanshan Yu, Jinxiang Wang, Fang Yang, Haibo Luo, Lijuan Yu

**Affiliations:** 1Agro-Products Processing Research Institute, Yunnan Academy of Agricultural Sciences, Kunming 650221, China; shenhui@yaas.org.cn (H.S.); wangyan9612@163.com (Y.W.); shipingpingtcm@163.com (P.S.); ynveg@163.com (H.L.); ncpjgskgk1215@163.com (F.Y.); 2College of Food Science and Technology, Yunnan Agricultural University, Kunming 650201, China; 3School of Food Science and Pharmaceutical Engineering, Nanjing Normal University, Nanjing 210023, China; chenyn813@163.com; 4Sericultural & Agri-Food Research Institute, Guangdong Academy of Agricultural Sciences, Guangzhou 510610, China; hu.tenggen@foxmail.com (T.H.); yuyuanshan1@163.com (Y.Y.); 5School of Life Sciences, Datong University, Datong 037054, China; wangjx11@163.com

**Keywords:** antifungal capacity, antioxidant capacity, bamboo leaf, genotype, volatile profile

## Abstract

Bamboo leaves contain high concentrations of various biologically active compounds, such as polyphenols and volatiles, making them attractive as raw resources for antioxidant additives in the food industry. Here, we investigated the total phenolic content (TPC) and total flavonoid content (TFC) of four bamboo leaf extracts from two species (*Phyllostachys edulis* and *Chimonocalamus delicatus*) at two growth stages (first and second years). Antioxidant capacity was determined based on the radical-scavenging capacity against 2,2-diphenyl-1-picrylhydrazyl (DPPH) and 2,2′-azinobis(3-ethylbenzothiazoline-6-sulfonic acid) (ABTS^+^). We also assessed the antifungal capacity based on mycelial growth inhibition of *Colletotrichum musae* (*C. musae*), *Botrytis cinerea* (*B. cinereain*), and *Alternaria alternata* (*A. alternata*). Pearson’s correlation coefficients showed that the TPC was significantly (*p* < 0.01) negatively correlated with the half-maximal inhibitory concentrations against DPPH and ABTS^+^, whereas the TFC was positively correlated with *C. musae* and *B. cinereain* growth inhibition, which suggest that TPC and TFC might be the major contributors to the antioxidant and antifungal capacities of bamboo leaves, respectively. The volatile organic compounds (VOCs) of bamboo leaves were also analyzed using gas chromatography–ion mobility spectrometry. The VOCs included twenty-four aldehydes, eleven alcohols, four furans, seven esters, fifteen terpenes, three ketones, one pyrazine, and thirty unidentified compounds. Principal component analysis, partial least squares discriminant analysis, and hierarchical cluster analysis were performed to assess the differences in the volatile profiles of the four bamboo leaf samples, from which 23 discriminatory VOCs with variable importance in the projection values > 1 were screened, and part of them were impacted by species or growth stage. These findings provide a theoretical foundation for the use of bamboo leaves.

## 1. Introduction

Bamboos are perennial, evergreen plant species in the Gramineae family and represent abundant forest resources with high economic and ecological value [1]. China has one of the largest areas of bamboo forest worldwide, accounting for 20% of the total area of bamboo globally [2]. Moso bamboo (*Phyllostachys edulis* [Carrière] J. Houz, *P. edulis*) belongs to the Bambusoideae (*Poaceae*) subfamily and is distributed widely in China, accounting for 73% of the total bamboo area [3]. *P. edulis* is characterized by high growth and reproductive rates, and has various uses in traditional medicines, building materials, and handicrafts. *Chimonocalamus delicatus* Hsueh Yi (*C. delicatus*) belongs to the Bambusoideae (*Chimonocalamus*) subfamily and is distributed in China, India, Bhutan, Vietnam, Thailand, and Myanmar; Yunnan, China represents the diversity center of this species [4]. The inner cavity of most species in the *Chimonocalamus* subfamily contains phenolic and flavonoid compounds, which confer its biologically active properties. However, some species in this subfamily are narrowly distributed and are decreasing mainly due to habitat destruction; therefore, they are listed as rare and protected species [4].

Bamboo leaves are usually discarded as byproducts, wasting potential resources and causing pollution. However, they are attracting growing attention as a source of phenolic acids, flavonoids, volatile oils, polysaccharides, and amino acids, making them promising sources of natural biologically active compounds [5,6]. Bamboo leaves exhibit high antioxidant, anti-inflammatory, and antiviral capacities. Bamboo leaf extract (BLE) has been included in the Chinese National Standard GB2760-2014 (revised from GB2760-2003) National Food Safety Standard for Uses of Food Additives as a legal additive in food production due to its potent antioxidant capacity [7]. Gong et al. [8] found that the n-butanol fraction of BLE displayed the highest total phenolic content (TPC) and total flavonoid content (TFC), and showed corresponding high radical-scavenging capacities compared to the water fraction of BLE and bamboo shavings extracts. Zhang et al. [9] found that BLE showed therapeutic effects against cardiac fibrosis in diabetic patients by inhibiting inflammation, oxidative stress, and apoptosis. Ma et al. [10] identified seven compounds in bamboo leaves using high-performance liquid chromatography and found that *Phyllostachys nigra* contained the highest isoorientin and isovitexin contents, *Lophatherum gracile* contained the highest cynaroside content, and *Pleioblastus amarus* contained the highest orientin content, which were related to their antioxidant capacities.

Bamboo leaves also have been reported to possess antimicrobial capacity [5,11]. Postharvest diseases caused by pathogenic fungi are the major cause of postharvest decay in fruits and vegetables. Chemical fungicides are commonly used to protect fruits and vegetables from postharvest diseases; however, they can have unintended consequences, such as increased antibiotic resistance, environmental pollution, and damage to human health. Thus, the development of natural antimicrobial agents to replace chemical fungicides has been a focal point of research. *Colletotrichum musae* (*C. musae*), *Botrytis cinerea* (*B. cinerea*), and *Alternaria alternate* (*A. alternata*) are major pathogenic fungi found on fruits and vegetables during preservation; infection by these fungi can deteriorate the quality of fruits and vegetables, causing large economic losses [12,13,14]. Liao et al. [15] found that *P. edulis* leaf extract significantly prevented and controlled phytophthora blight in pepper due to its prominent antifungal capacities against various species, including *Phytophthora capsici*, *Fusarium graminearum*, *Valsa mali*, *Botryosphaeria dothidea*, *Venturia nashicola*, and *B. cinerea*. While bamboo leaves may show promise as raw materials for developing broad-spectrum bactericides or fungicide adjuvants, few studies have investigated their antimicrobial effects.

Volatile organic compounds (VOCs) are potential sources of not only pleasant odors and biological activities but also important characteristics in different species and growth stages. However, there is little information on the VOCs in bamboo leaves, which constrains their use. Technologies including gas chromatography–olfactory–mass spectrometry (GC-O-MS), gas chromatography–mass spectrometry (GC-MS), and gas chromatography–ion mobility spectrometry (GC-IMS) have been widely applied to separate and sensitively detect VOCs [16,17]. For instance, Takahashi et al. [18] identified 89 compounds in *P. edulis* stems via GC and GC-MS, and determined the most potent odorants through dilution analysis. Mi et al. [19] screened 24 VOCs with variable importance in the projection (VIP) values > 1 as the discriminant compounds of Korean and Jize chili peppers by GC-IMS. Shen et al. [20] analyzed the odor-active compounds in *P. edulis* leaf, stem, and powder using GC-IMS, and further characterized 39 and 59 main odor-active compounds in *P. edulis* leaves using GC-O-MS and GC × GC-O-MS, respectively. These studies highlight the feasibility of elucidating critical and discriminative VOCs in bamboo leaves.

In this study, we assessed the biological activities and related compounds in four types of bamboo leaves from two species (*P. edulis* and *C. delicatus*) and two growth stages (first- and second-year leaves). First, we analyzed and compared their TPC and TFC. Then the radical-scavenging capacities and antifungal capacities against *C. gloeosporioides*, *B. cinerea*, and *A. alternata* of each bamboo type in vitro were determined. In addition, we identified correlations between the antioxidant and antifungal capacities and TPC and TFC of the bamboo leaves. Using GC-IMS, we analyzed the VOCs in the leaves, and multivariate statistical analyses were performed to identify the critical and discriminative VOCs (VIP > 1). Our results provide a theoretical foundation to promote the use of bamboo leaves.

## 2. Materials and Methods

### 2.1. Chemicals and Reagents

The following chemicals and reagents were purchased from Shanghai Yuanye Bio-Technology Co., Ltd. (Shanghai, China): Folin–Ciocalteu reagent, 2,2-diphenyl-1-picrylhydrazyl (DPPH), aluminum nitrate, 2,2′-azinobis(3-ethylbenzothiazoline-6-sulfonic acid) (ABTS), sodium carbonate, sodium nitrite, aluminum chloride, sodium hydroxide, ascorbic acid, gallic acid, rutin, and Trolox. Analytical grade ethanol was purchased from Sichuan Xilong Science Co., Ltd. (Chengdu, China). Pure water was prepared using a secondary reverse osmosis system (JYS-500L; Shanghai Juyuan Automation Technology Co., Ltd., Shanghai, China).

### 2.2. Materials

#### 2.2.1. Bamboo Leaves

The annual and biennial leaves of *P. edulis* and *C. delicatus* were harvested from Yunnan Zhenzhu Agricultural Technology Co., Ltd. (Kunming, China). (upper left longitude: 103.088665°, upper left latitude: 25.337019°; lower left longitude: 103.089867°, lower left latitude: 25.333840°; upper right longitude: 103.092313°, upper right latitude: 25.336980°; lower right longitude: 103.091948°, lower right latitude: 25.334829°) in April 2022 as the experimental materials (Figure 1). The samples were air-dried and pulverized (20 mesh), vacuum packed, and frozen at −20 °C until analysis.

#### 2.2.2. Fungal Strains

The *C. musae* strain was obtained from BeNa Culture Collection (Beijing, China). *B. ci-nerea* and *A. alternata* strains were obtained from China Center for Type Culture Collec-tion (Wuhan, China). The three fungal strains and their derivatives were cultured on potato dextrose agar (PDA) medium (200 g L^−1^ potato, 20 g L^−1^ glucose, 20 g L^−1^ agar) at 28 °C to test the antifungal capacities of the BLEs.

### 2.3. BLE Preparation

Samples of dried bamboo leaf powder were extracted twice with aqueous ethanol (60:40, ethanol:water, *v*/*v*). The supernatants of the two extractions were combined, filtered, evaporated at 60 °C, and dried at 60 °C for subsequent analysis.

### 2.4. Total Phenolic Content (TPC) Determination

TPC was determined using the Folin–Ciocalteu reagent colorimetric method according to Shang et al. [21], with modifications. First, 0.05 g of BLE was accurately weighed and reconstituted to a volume of 10 mL using aqueous ethanol (60:40, ethanol:water, *v*/*v*). Then, 0.1 mL of each sample was added to a solution of 0.5 mL of Folin–Ciocalteu reagent and 0.4 mL of water for 8 min. Next, a mixture of 1.5 mL of aqueous sodium carbonate (20:80, sodium carbonate:water, *w*/*v*) and 0.8 mL of water was added and incubated for 90 min. Finally, the absorbance at 760 nm was measured. The final values were expressed in milligrams of gallic acid per gram of dry bamboo leaf sample.

### 2.5. Total Flavonoid Content (TFC) Determination

TFC was measured using the aluminum chloride colorimetric method reported by Shang et al. [21], with modifications. The samples were initially prepared following the same process described in Section 2.4. Then, each sample solution (0.1 mL) was mixed with 0.15 mL of ethanol and 0.15 mL of aqueous aluminum chloride (20 g L^−1^) for 5 min. The absorbance at 430 nm was measured. The final values were expressed in milligrams of rutin per gram of dry bamboo leaf sample.

### 2.6. Antioxidant Capacity Determination

#### 2.6.1. DPPH Radical-Scavenging Capacity

To assess the DPPH radical-scavenging capacity, we followed the method of Ma et al. [22], with modifications. In brief, a 100 µL aliquot sample was mixed with 100 µL of DPPH radical working solution in ethanol. The absorbance at 734 nm was measured. The scavenging rate was calculated as follows:(1)$$DPPH\;radical\;scavenging\;\left(\%\right)=[(A_{0}-A_{1})/A_{0}]\times 100$$
where *A*_0_ is the absorbance of the DPPH solution and *A*_1_ is the absorbance of the sample.

#### 2.6.2. ABTS^+^ Radical-Scavenging Capacity

The ABTS^+^ radical-scavenging capacity was assessed according to the method of Tundis et al. [23], with modifications. ABTS^+^ was generated by freshly reacting ABTS (7 mM) with K_2_S_2_O_8_ (2.45 mM). The mixture was left at room temperature in the dark for 12 h, and diluted with ethanol to an absorbance of 0.70 ± 0.20. After mixing 1.2 mL of the ABTS^+^ solution and 0.3 mL of the sample solution, the solution was incubated at room temperature for 6 min. Absorbance was measured at 734 nm. The scavenging rate was calculated as follows:(2)$${ABTS}^{+}\;scavenging\;(\%)=\left[\left(A_{0}-A_{1}\right)/A_{0}\right]\times 100$$
where *A*_0_ is the absorbance of the control and *A*_1_ is the absorbance of the sample.

### 2.7. Antifungal Capacity Assays

Three pathogenic fungi (*C. musae*, *B. cinerea*, and *A. alternata*) stored in aqueous glycerol (30:70, glycerol:water, *v*/*v*) at −80 °C were regenerated and activated before use. Each activated species was grown on PDA in culture dishes at 28 °C for 7 days, and the resultant fungus cake was passed through a 7 mm hole punch. Aliquots of each BLE sample solution were added to PDA medium at working concentrations of 5.0 mg mL^−1^, with an equivalent volume of aqueous ethanol (60:40, ethanol:water, *v*/*v*) as the control. Then, the cakes were inversely set on the PDA media containing BLE (or control) and incubated at 28 °C. Observations were made after 2, 4, and 6 days. The antifungal capacities were assessed based on the mycelial growth diameters and calculated as follows:(3)$$Inhibition\;rate\left(\%\right)=\left[1-\frac{Control\;diamenter-Sample\;diameter}{Control\;diameter}\right]\times 100\%$$

### 2.8. GC-IMS Assay

The volatile profiles of the four types of bamboo leaves were analyzed using a FlavourSpec^®^ GC-IMS system (G.A.S Company, Berlin, Germany) as described by Shen et al. [20], with adjustments. Fresh bamboo leaves were ground, and a 2 g aliquot was transferred to a headspace bottle (20 mL) and incubated at 60 °C for 20 min. Next, 200 µL of the sample solution was automatically injected using a heated syringe needle (85 °C) before incubating at 60 °C for 20 min at 500 r min^−1^. VOCs were separated through an MXT-5 column (15 m × 0.53 mm × 1 µm; Restek, Centre County, PA, USA). The chromatographic column was maintained at 60 °C and the running time was 30 min with an IMS temperature maintained at 45 °C; nitrogen was used as both the drift and carrier gases. The flow rate of the carrier gas was varied as follows: 2 mL min^−1^ during 0–2 min, 10 mL min^−1^ during 2–10 min, 100 mL min^−1^ during 10–20 min, 150 mL min^−1^ during 20–25 min, and 150 mL min^−1^ during 25–45 min.

### 2.9. Statistical Analysis

The GC-IMS data were processed and viewed using the GC-IMS instrument analysis software (VOCal 0.1.3), which included a laboratory analytical viewer, two plug-ins (Reporter and Gallery Plot), and GC × IMS Library Search. The Reporter plug-in was used to visualize the two-dimensional spectra to compare the spectral differences among samples. The Gallery Plot plug-in was used to visualize the VOC fingerprints to compare differences among samples. The qualitative analysis of VOCs was based on the NIST database and IMS database built in the GC × IMS Library. Principal component analysis (PCA) and Pearson’s correlation analysis were performed and plotted with OriginPro 2021 (OriginLab, Northampton, MA, USA). Partial least squares discriminative analysis (PLS-DA), permutation validation of the PLS-DA model, hierarchical cluster analysis (HCA), and VIP analysis were performed using the software SIMCA ver. 14.1 (Sartorius, Göttingen, Germany). The VIP data were plotted in GraphPad Prism ver. 5.0 (GraphPad, Boston, MA, USA). The heatmap was drawn with OriginPro 2021. Analysis of variance and Duncan’s multiple range test were performed using SPSS ver. 23.0 (IBM Corp., Armonk, NY, USA). Data are expressed as the means ± standard deviations.

## 3. Results

### 3.1. TPC and TFC of BLE

Phenolics, including phenolic acids and flavonoids, might represent the constituents in bamboo leaves that confer their antioxidant and antifungal capacities [22,24]. Therefore, we determined the TPC and TFC in the four BLEs (Table 1). The TPC of *P. edulis* BLE was higher than that of *C. delicatus* BLE, whereas the TFC in *C. delicatus* BLE was higher than that in *P. edulis* BLE; the same trends were observed in first- and second-year leaves. Meanwhile, the TPC significantly increased with bamboo growth; compared to first-year leaves, the TPC in the second-year leaves of *P. edulis* and *C. delicatus* increased by 36% (to 898.90 mg g^−1^) and 11% (to 605.54 mg g^−1^), respectively. Similarity, the TFC increased from 86 mg g^−1^ in first-year leaves to 93 mg g^−1^ in second-year leaves of *P. edulis*, and from 110.75 mg g^−1^ to 135.60 mg g^−1^ in *C. delicatus*. The changes in the TPC and TFC with growth might be the result of changes in biosynthesis and the transformation of secondary metabolites in plants [19,25].

Interestingly, the TPCs and TFCs in *P. edulis* and *C. delicatus* were much higher than those in *P. nigra* of 26.82 mg g^−1^ and 6.57 mg g^−1^, respectively, when extracted following a pressurized liquid method under optimal conditions [21]. In addition, the TFCs in *Phyllostachys pubescens* leaves were somewhat higher than the yields obtained using Soxhlet extraction (85.6 mg g^−1^), reflux extraction (77.4 mg g^−1^), ultrasonic extraction (42.4 mg g^−1^), and the homogenate-assisted vacuum-powered bubble method (76.1 mg g^−1^) [26]. The disparities in the TPCs and TFCs of BLEs between our and previous studies might have resulted from the extraction method, species, and bamboo growth conditions. To further clarify the antioxidant and antifungal mechanisms of bamboo leaves, further research is needed to isolate individual antioxidants, confirm their chemical structures, and identify structure–capacity relationships.

### 3.2. Antioxidant Capacities against DPPH and ABTS^+^ Radicals

Bamboo leaves are used widely in traditional Chinese medicine, with early records in Sheng Nong’s Herbal Classic Collections (East Han Dynasty) and the Compendium of Materia Medica (Ming Dynasty) [6]. With the development of modern medicine and food technologies, bamboo leaves have been confirmed to have various biological activities, such as antioxidant, anti-inflammatory, antiviral capacities [20].

We first assessed the antioxidant capacity of four BLE samples by determining the half-maximal inhibitory concentration (IC_50_) against DPPH and ABTS^+^ radicals in vitro. The results are shown in Table 2, where L-ascorbic acid serves as a positive control. Based on the IC50 values, the DPPH radical-scavenging capacity followed the order of second-year *P. edulis* BLE > first-year *P. edulis* BLE > second-year *C. delicatus* BLE > first-year *C. delicatus* BLE. The IC_50_ of second-year *P. edulis* BLE against DPPH (0.63 mg mL^−1^) was markedly lower than that of first-year *P. edulis* BLE (0.81 mg mL^−1^). Additionally, the IC_50_ of second-year *C. delicatus* against DPPH (0.84 mg mL^−1^) was lower than that of first-year *C. delicatus* (0.89 mg mL^−1^). The BLE samples showed similar trends in their ABTS^+^ radical-scavenging capacities, although the differences between *P. edulis* and *C. delicatus* and between first- and second-year leaves were smaller than those for DPPH. Overall, *P. edulis* exhibited higher antioxidant capacities than *C. delicatus*, and second-year leaves had stronger antioxidant capacities than first-year leaves. Thus, the antioxidant capacities varied by species and increased with bamboo growth.

### 3.3. Antifungal Capacities against C. musae, B. cinerea, and A. alternata

Studies have investigated various biological activities of BLE, including the antioxidant capacity, anti-inflammatory capacity, and the prevention of cardiovascular and metabolic diseases [5,6]; however, little is known about the antifungal capacity of BLE, or its potential in the preservation of fruits and vegetables.

Here, we evaluated the inhibitory effect of BLE on *C. musae*, *B. cinerea*, and *A. alternata* mycelial growth in vitro (Figure 2). The four types of BLEs inhibited the mycelial growth of each species to varying degrees. Second-year *C. delicatus* BLE had the highest inhibition rates against *C. musae* and *B. cinerea*, which were significantly higher (*p* < 0.05) than those of second-year *P. edulis* BLE, with inhibition rates of 19.17% and 35.70%, respectively, after 6 days of incubation. Overall, second-year leaves had greater inhibitory effects against *C. musae* and *B. cinerea* than first-year leaves. All four BLE samples significantly inhibited *A. alternata* growth by 42.74%, 43.08%, 41.65%, and 42.54%, respectively, with no marked differences in the rates. The BLE samples displayed higher inhibition of *A. alternata* than *C. musae* and *B. cinerea*.

Overall, the *C. delicatus* BLEs exhibited higher antifungal capacities than the *P. edulis* BLEs, and second-year BLEs showed stronger capacities than first-year BLEs. Moreover, the BLEs exhibited antifungal capacities against all three tested fungi but with a higher inhibitory capacity against *A. alternata* than *C. musae* and *B. cinerea*. Our findings support those of Liao et al. [15], who reported that *P. edulis* leaf extract showed strong antifungal capacities against *P. capsica* (inhibitory rate: 100.00%) *F. graminearum* (75.12%), *V. mali* (60.66%), *B. dothidea* (57.24%), *V. nashicola* (44.62%), and *B. cinerea* (30.16%). Taken together, these results highlight the potential for using bamboo leaves to develop broad-spectrum bactericides or fungicide adjuvants.

### 3.4. Pearson’s Correlation Analysis of Biological Components and Capacities

Antioxidant and antifungal capacities are impacted by the abundance of phenolic acids and flavonoids [5,6,27,28]. Hence, we performed Pearson’s correlation analysis to identify relationships between biological activities (i.e., antioxidant and antifungal capacities) and the abundance of biological compounds (i.e., TPC and TFC) in bamboo leaf extracts (Figure 3). We observed significant negative correlations (*p* < 0.01) between the IC_50_ of TPC against DPPH (correlation coefficient: −1.00) and ABTS^+^ (−0.94). These results suggest that the TPC is a major contributor to the antioxidant capacity of bamboo leaves. In addition, the TFC was highly positively correlated (*p* < 0.01) with *C. musae* (0.90) and *B. cinerea* (0.81) growth inhibition. Thus, the TFC of bamboo leaves might be important in conferring the inhibitory capacity against *C. musae* and *B. cinerea*. Additionally, a higher TPC was detected in *C. delicatus* leaves (Table 1), which might explain why the antifungal activities of *C. delicatus* were stronger than *P. edulis* leaves. Meanwhile, *A. alternata* growth inhibition was weakly correlated with both the TFC (−0.23) and TPC (0.51).

### 3.5. Volatile Profile of Bamboo Leaves Using GC-IMS

From the analysis in Section 3.2 and Section 3.3, *P. edulis* leaves exhibited a greater antioxidant capacity, but *C. delicatus* leaves showed a stronger antifungal capacity. In addition, second-year bamboo leaves showed stronger biological activities than first-year leaves. Thus, we analyzed their volatile profiles using GC-IMS. The results were visualized using a two-dimensional chromatogram and a fingerprint (Figure 4). As shown in Figure 4a, *C. delicatus* contained a greater diversity of VOCs than *P. edulis*, and the biennial bamboo leaves of both species contained more diverse VOCs with higher concentrations than annual leaves.

In addition, more comprehensive and intuitive information is displayed in Figure 4b. In this fingerprint, 95 VOCs are displayed; each column represents the VOCs observed in each bamboo leaf sample, and each row represents the signal peak of a certain VOC labeled with the name or a number (unidentified), in which the deeper color represents a higher abundance of the VOCs from low to high, with pure black representing the concentration of a substance close to zero, as described by Zhou et al. (2022) [29]. Figure 4b is divided into 4 regions. The components in region A were present in *P. edulis* and *C. delicatus* at both growth stages. The components in region B were dominant in *P. eduli*. Additionally, region B was further divided into 3 sub-regions; B1 includes the components that were dominant both in annual and biennial *P. edulis*; B2 includes the components that were dominant in biennial rather than annual *P. edulis*; and the components in B3 were converse in annual and biennial *P. edulis*. The components in region C were dominant in *C. delicates*, which is also divided into 3 sub-regions; C1 includes the components that were dominant in annual *C. delicatus*; C2 includes the components that were dominant in both annual and biennial *C. delicatus*; and C3 includes the components that were dominant in biennial *C. delicatus*. Region D includes VOCs that could be traced both in *P. edulis* and *C. delicatus*, although with low concentrations.

### 3.6. Qualitative Analysis of VOCs in Bamboo Leaves

We calculated the relative content (RC) of each class based on its corresponding normalized peak area (Table 3). Aldehydes cumulatively accounted for 38.34% of all VOCs in bamboo leaves, followed by alcohols (11.21%), furans (10.86%), esters (3.92%), terpenes (3.62%), ketones (2.35%), pyrazine (1.80%), and unknown VOCs (27.90%).

Aldehydes are mainly derived from unsaturated fatty acid oxidation, and many have a pleasant odor with a low odor threshold. In this study, 10 aldehydes had RCs higher than 1% (Table 3). Shen et al. [20] also reported that aldehydes had the highest RC in *P. edulis*, and speculated that this might be beneficial for developing natural food additives from *P. edulis* leaves. Additionally, among them, (E)-2-pentenal [20] and 2-methylbutanal and 3-methylbutanal [30] may represent markers of microbial capacity spoilage based on amino acid degradation. Moreover, although its RC was lower than 1%, the importance of nonanal is worth exploring. Zhang et al. [31] reported that nonanal dose-dependently inhibited *Penicillium cyclopium* mycelial growth by disrupting the integrity of the fungal cell membrane.

Alcohols were the second-most dominant VOCs in bamboo leaves, and three alcohols had RCs higher than 1%: 1-propanol-M, 2-butanol-D, and 1-propanethiol-D. Among them, 1-propanol is pungent and bitter [29], and is found in many plant species, including *P. edulis* [20]. Meanwhile, 2-butanol is used as a marker of potato infection by *Pectobacterium carotovorum* [32].

While only four furans were detected (2-ethylfuran-M, 2,5-dimethylfuran, 2-ethylfuran-D, and 2-pentylfuran), all four had relatively high RCs. For instance, the cumulative RC (CRC) of 2-ethylfuran (which included two structures) was 6.05%, followed by 2,5-dimethylfuran with an RC of 4.56%. 2-Ethylfuran is produced from lipid degradation of linoleic acid [20] and contributes to grassy, fatty, and sweet fruity aromas [33]. By contrast, many terpenes were identified, but the RC of each was relatively low. Among terpenes, 2-heptanone-D had the highest RC of 0.66%.

Finally, ethyl butyrate and 2,3-pentadione were the only compounds with RCs higher than 1% among esters and ketones, respectively. Ethyl butyrate has an aroma similar to kiwifruit and pineapple [34], whereas 2,3-pentadione is produced via the Maillard reaction and has a buttery and caramel odor [35].

The volatile profiles of bamboo leaves are complex, and analytical results may be influenced by the instruments, analytical methods, and conditions used, as well as the information available in VOC databases. We detected 30 unidentified compounds, with a cumulative RC of 27.90%; among them, the cumulative RC of the eight dominant VOCs was 17.17% (Appendix A). Given the high proportion of unidentified compounds, future studies should aim to expand VOC databases.

### 3.7. Multivariate Statistical Analysis of VOCs

#### 3.7.1. PCA

To identify the VOCs most influenced by the bamboo species and growth stage, we performed a PCA of 65 VOCs (Figure 5, Table 3). 

The first two principal components (PCs) explained 38.1% and 32.3% of the total variance, respectively, which implies that these two PCs could explain most differences in the volatile profiles of the four bamboo leaf samples. The four samples were all well separated: second-year *P. edulis* and second-year *C. delicatus* had negative scores on PC1 and PC2; first-year *P. edulis* had positive scores on PC1 and PC2; and first-year *C. delicatus* had a positive score on PC1 and negative score on PC2. The distances between samples represent their degree of difference or similarity. For instance, the VOCs in second-year *C. delicatus* and second-year *P. edulis* leaves were relatively close, whereas those in first-year *C. delicatus* leaves were relatively far from first-year *P. edulis* leaves; thus, the VOCs in second-year leaves were more similar between species than those in first-year leaves. Overall, the PCA revealed that bamboo leaves from different species and growth stages could be well distinguished according to their volatile profiles.

#### 3.7.2. Multivariate Analysis of VOCs

To confirm the PCA results, we performed PLS-DA (Figure 6a). As important parameters in PLS-DA models, R^2^X, R^2^Y, and Q^2^ values close to 1 indicate that the model is more stable and reliable [36]. In this study, R^2^X, R^2^Y, and Q^2^ were 0.985, 0.997, and 0.987, respectively, indicating that the model was sufficiently reliable for further analysis. Next, a permutation test with 200 responses was performed to verify the reliability of the PLS-DA model (Figure 6b). The results of R^2^ = 0.294 and Q^2^ = −0.495 confirmed that the PLS-DA model was highly reliable and exhibited no overfitting. In addition, HCA was applied (Figure 6c); the clustering of the samples was consistent with the PCA and PLS-DA results. Moreover, VIP score was calculated and 23 VOCs with VIP values > 1 were shown in Figure 6d, including twelve aldehydes, three alcohols, four furans, three esters, and one terpene. Among these 23 compounds, 14 VOCs were also detected in *P. edulis* leaves by Shen et al. [20], including (E)-2-pentenal-M, 3-methylbutanal-D, 2-methylbutanal, and 3-methylbutanal-M.

### 3.8. Distinguishing VOCs in Bamboo Leaves by Species and Growth Stage

To identify distinguishing features of the four types of BLEs, the 23 compounds were further clustered according to their RC in each sample (Figure 7). There were four, two, one, and three dominant discriminative compounds in first-year *P. edulis* (1-propanethiol-D, 1-propanol-M, 3-methylbutanal-M, and (E,E)-2,4-heptadienal), second-year *P. edulis* (propanal-D and ethyl butyrate), first-year *C. delicatus* (2-heptanone-D), and second-year *C. delicatus* (butanal-D, 2-phenyl-1,3-dioxolane-4-methanol, and nonanal-M), respectively. The abundances of some VOCs were impacted by the species or growth stage. For instance, the abundances of methyl-2-furoate and 2,3-dimethylpyrazine were significantly higher in *C. delicatus* than *P. edulis*, and no VOCs were more abundant in *P. edulis* than *C. delicatus* (Figure 7, Table 4). Methyl-2-furoate and 2,3-dimethylpyrazine had VIP values of 1.11 and 1.70, respectively, suggesting that they may be effective for distinguishing *P. edulis* and *C. delicatus* leaves. In addition, the abundances of (E)-2-pentenal-M, (E,E)-2,4-hexadienal-D, 2,5-dimethylfuran, and propanal-M were higher in first-year leaves than in second-year leaves. By contrast, five VOCs had higher concentrations in second-year leaves: ethyl 3-hydroxybutanoate, 2-methylpropanal, 2-ethylfuran-D, 2-methylbutanal, and 3-methylbutanal-D. Among them, 2,5-dimethylfuran and 2-ethylfuran-D had the highest VIP values of 2.42 and 2.14, respectively, suggesting that they may be the best markers for differentiating first- and second-year bamboo leaves.

## 4. Conclusions

We compared the TPC, TFC, and antioxidant and antifungal capacities of the BLEs of two bamboo species (*P. edulis* and *C. delicatus*) at two growth stages (first- and second-year leaves). Second-year *P. edulis* had the highest TPC, whereas second-year *C. delicatus* had the highest TFC. *P. edulis* exhibited higher antioxidant capacities, whereas *C. delicatus* possessed stronger antifungal capacities. Moreover, second-year bamboo leaves showed stronger biological activities than first-year leaves. In addition, we detected 95 VOCs in bamboo leaves using GC-IMS, mainly aldehydes, alcohols, furans, esters, terpenes, and ketones. PCA, PLS-DA, and HCA of the normalized peak areas of VOCs clearly distinguished the four BLE samples by species and growth stage. Furthermore, 23 characteristic VOCs (VIP > 1) were identified to further discriminate the four BLE samples. Our findings provide theoretical information for the development of natural antioxidants and preservatives derived from bamboo leaves, which may be of use in the food industry as fruit and vegetable preservatives.

## Figures and Tables

**Figure 1 foods-13-00480-f001:**
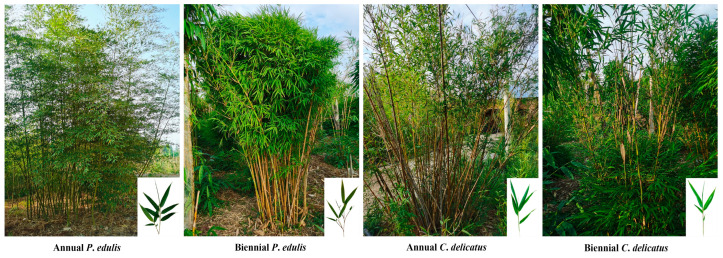
Annual and biennial leaves of *P*. *edulis* and *C*. *delicatus*.

**Figure 2 foods-13-00480-f002:**
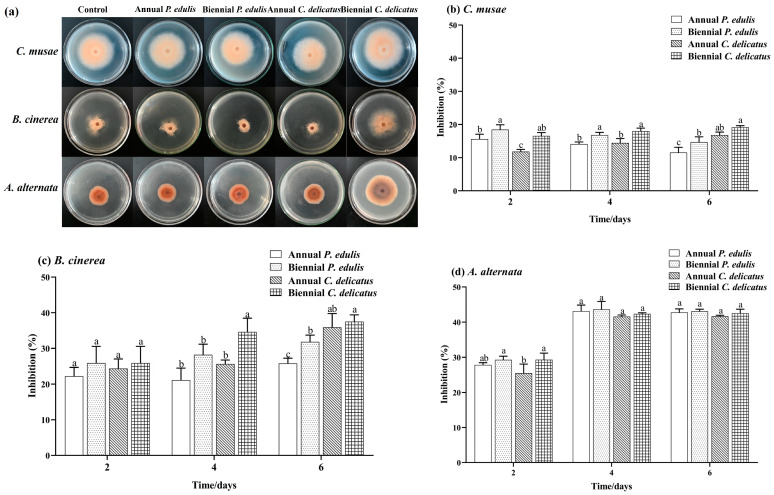
Antifungal capacities of BLE. (**a**) Mycelia grown on PDA without (control) or with BLE (5.0 mg mL^−1^); inhibition of (**b**) *C. musae*, (**c**) *B. cinerea*, and (**d**) *A. alternata*. Error bars indicate the standard deviations of triplicate experiments, and lowercase letters represent significant differences (*p* < 0.05).

**Figure 3 foods-13-00480-f003:**
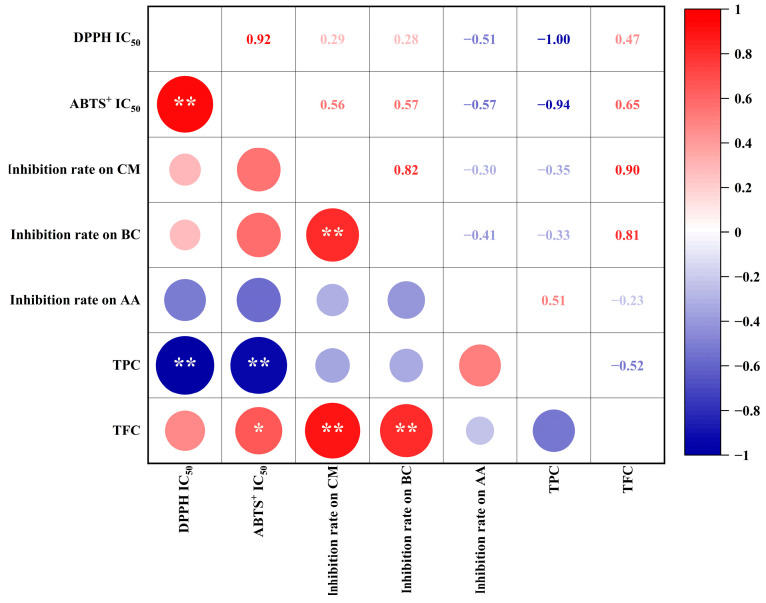
The Pearson’s correlations between TPC, TFC and antioxidant and antifungal capacities of BLE. * indicates a significance level of *p* < 0.05; ** indicates extremely significant at *p* < 0.01.

**Figure 4 foods-13-00480-f004:**
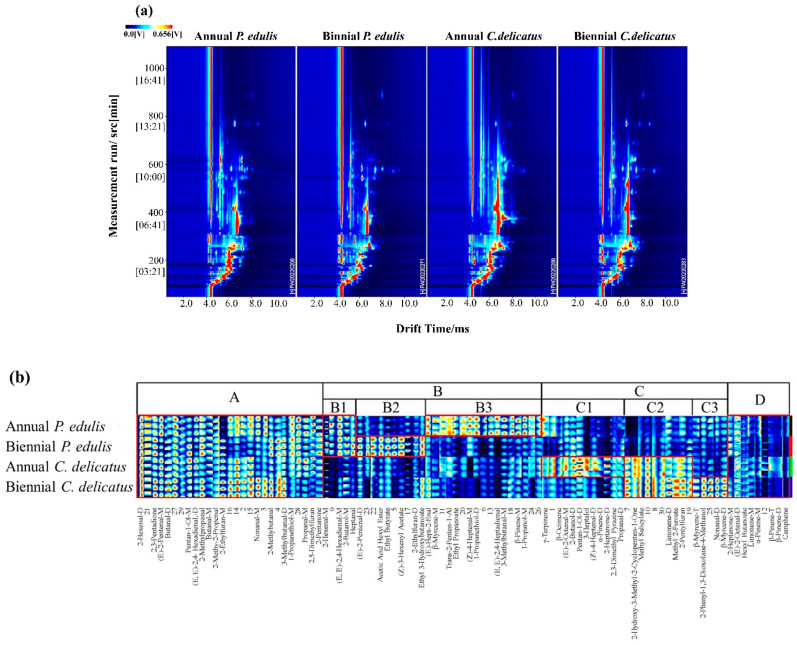
Volatile profiles of bamboo leaves identified using GC-IMS. (**a**) Two-dimensional spectrum, where the x-axis represents the drift time of IMS, the y-axis represents the retention time of GC, and the red vertical line indicates the RIP; (**b**) characteristic fingerprint, where each column represents the signal peak of the same VOC measured in triplicate, and the unidentified substances are described by numbers.

**Figure 5 foods-13-00480-f005:**
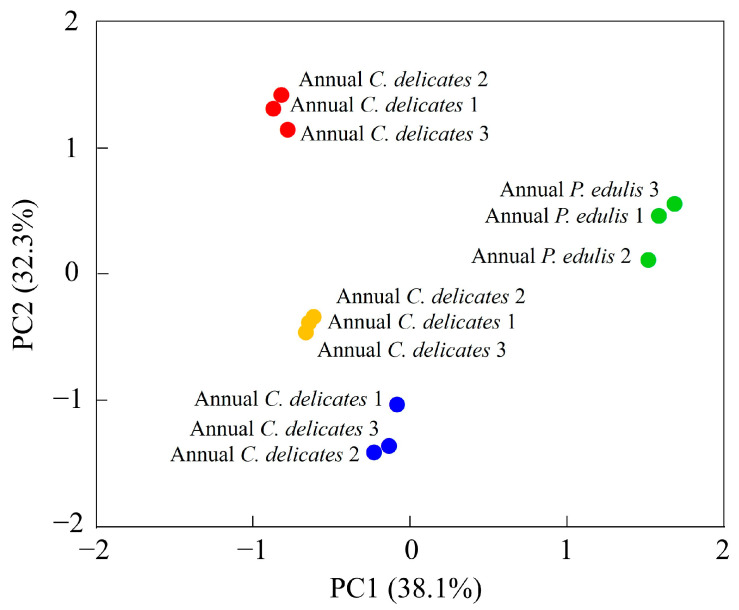
PCA of VOCs in bamboo leaves detected using GC-IMS.

**Figure 6 foods-13-00480-f006:**
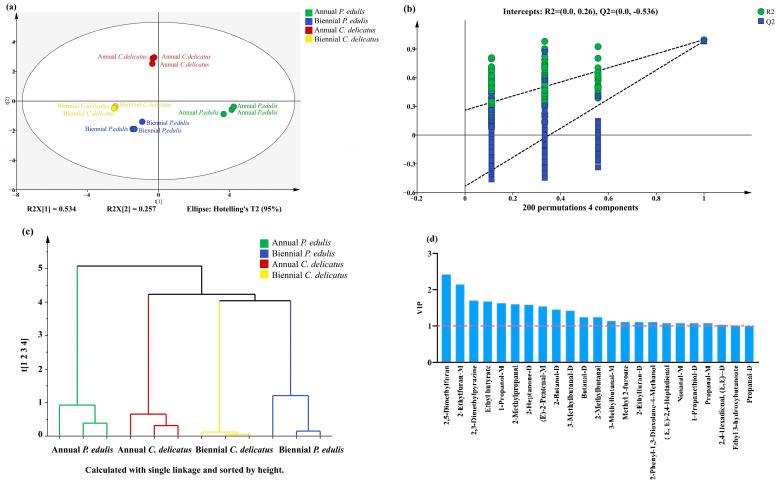
Multivariate statistical analysis of VOCs in bamboo leaves. (**a**) PLS-DA score plot; (**b**) permutation test result of PLS-DA (200 responses); (**c**) HCA result; and (**d**) 23 VOCs with VIP values > 1.

**Figure 7 foods-13-00480-f007:**
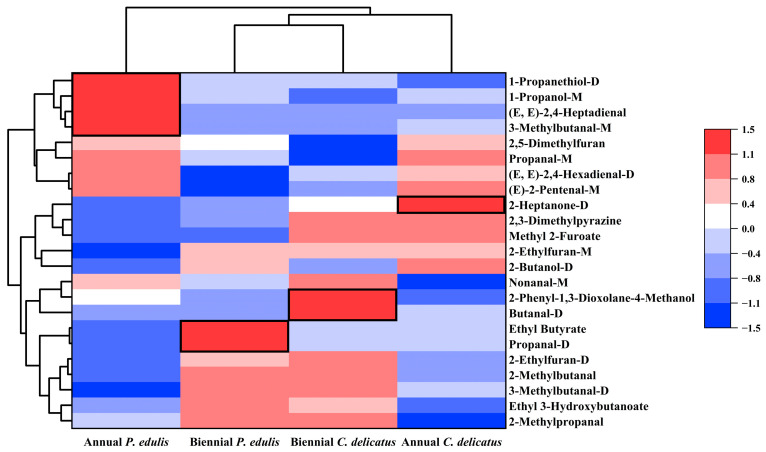
Multivariate statistical analysis of VOCs in bamboo leaves.

**Table 1 foods-13-00480-t001:** The total phenolic and total flavonoid contents of BLE.

Sample	TPC (mg GAE g^−1^)	TFC (mg RT g^−1^)
Annual *P. edulis*	662.94 ± 0.81 ^b^	86.00 ± 0.06 ^d^
Biennial *P. edulis*	898.90 ± 0.81 ^a^	93.00 ± 0.06 ^c^
Annual *C. delicates*	546.73 ± 0.81 ^d^	110.75 ± 0.06 ^b^
Biennial *C. delicatus*	605.54 ± 1.07 ^c^	135.60 ± 0.06 ^a^

Data are expressed as means ± SD. The superscripted small letter in each column indicates a significant difference (*p* < 0.05, *n* = 3). GAE, gallic acid; RT, rutin.

**Table 2 foods-13-00480-t002:** The in vitro antioxidant capacities of BLE.

Sample	IC_50_ Value for the DPPH Radical (mg mL^−1^)	IC_50_ Value for ABTS^+^ (mg mL^−1^)
Annual *P. edulis*	0.81 ± 0.00 ^c^	0.40 ± 0.00 ^c^
Biennial *P. edulis*	0.63 ± 0.00 ^d^	0.32 ± 0.00 ^d^
Annual *C. delicates*	0.89 ± 0.00 ^a^	0.57 ± 0.00 ^a^
Biennial *C. delicates*	0.84 ± 0.00 ^b^	0.50 ± 0.00 ^b^
L-Ascorbic acid	0.03 ± 0.00 ^e^	0.02 ± 0.00 ^e^

Data are expressed as means ± SD. The small letter superscripted in each column indicates a significant difference (*p* < 0.05, *n* = 3).

**Table 3 foods-13-00480-t003:** All VOCs in bamboo leaves detected using GC-IMS.

No.	Compound	CAS#	Formula	MW	RI	RC (%)	CRC (%)
Aldehydes							
1	2-Hexenal-D	505-57-7	C_6_H_10_O	98.1	855.5	6.88	38.34
2	(E)-2-Pentenal-M	1576-87-0	C_5_H_8_O	84.1	754	5.89	
3	3-Methylbutanal-D	590-86-3	C_5_H_10_O	86.1	649	3.94	
4	2-Methylbutanal	96-17-3	C_5_H_10_O	86.1	670.4	3.66	
5	2-Methylpropanal	78-84-2	C_4_H_8_O	72.1	531	3.10	
6	(E, E)-2,4-Hexadienal-D	142-83-6	C_6_H_8_O	96.1	923.6	2.04	
7	Butanal-D	123-72-8	C_4_H_8_O	72.1	575.3	1.99	
8	3-Methylbutanal-M	590-86-3	C_5_H_10_O	86.1	651.3	1.65	
9	Propanal-M	123-38-6	C_3_H_6_O	58.1	501.7	1.48	
10	2-Methyl-2-Propenal	78-85-3	C_4_H_6_O	70.1	562.6	1.20	
11	Propanal-D	123-38-6	C_3_H_6_O	58.1	493.5	0.95	
12	2-Hexenal-M	505-57-7	C_6_H_10_O	98.1	852.3	0.92	
13	(E)-2-Pentenal-D	1576-87-0	C_5_H_8_O	84.1	756.6	0.82	
14	Nonanal-M	124-19-6	C_9_H_18_O	142.2	1105.7	0.76	
15	2-Phenyl-1,3-Dioxolane-4-Methanol	1708-39-0	C_10_H_12_O_3_	180.2	975.9	0.56	
16	Trans-2-Penten-1-al	1576-87-0	C_5_H_8_O	84.1	731.7	0.46	
17	(E, E)-2,4-Hexadienal-M	142-83-6	C_6_H_8_O	96.1	923.1	0.44	
18	Butanal-M	123-72-8	C_4_H_8_O	72.1	590.9	0.34	
19	(E)-Hept-2-Enal	18829-55-5	C_7_H_12_O	112.2	961.9	0.34	
20	(E, E)-2,4-Heptadienal	4313-03-5	C_7_H_10_O	110.2	1012.1	0.32	
21	Heptanal	111-71-7	C_7_H_14_O	114.2	902	0.25	
22	Nonanal-D	124-19-6	C_9_H_18_O	142.2	1106.2	0.16	
23	(E)-2-Octenal-M	2548-87-0	C_8_H_14_O	126.2	1070.2	0.15	
24	(E)-2-Octenal-D	2548-87-0	C_8_H_14_O	126.2	1070.6	0.03	
Alcohols							
25	1-Propanol-M	71-23-8	C_3_H_8_O	60.1	566.6	3.39	11.21
26	2-Butanol-D	78-92-2	C_4_H_10_O	74.1	585	2.10	
27	1-Propanethiol-D	107-03-9	C_3_H_8_S	76.2	618.8	1.44	
28	1-Propanethiol-M	107-03-9	C_3_H_8_S	76.2	624.7	0.85	
29	2-Butanol-M	78-92-2	C_4_H_10_O	74.1	583.4	0.66	
30	Pentan-1-Ol-M	71-41-0	C_5_H_12_O	88.1	768.5	0.54	
31	3-Heptanol	589-82-2	C_7_H_16_O	116.2	895.4	0.54	
32	(Z)-4-Heptenal-D	6728-31-0	C_7_H_12_O	112.2	901.9	0.51	
33	Ethyl Propanoate	105-37-3	C_5_H_10_O_2_	102.1	708.8	0.47	
34	Pentan-1-Ol-D	71-41-0	C_5_H_12_O	88.1	767.5	0.45	
35	(Z)-4-Heptenal-M	6728-31-0	C_7_H_12_O	112.2	902.2	0.27	
Furans							
36	2-Ethylfuran-M	3208-16-0	C_6_H_8_O	96.1	706.8	4.96	10.86
37	2,5-Dimethylfuran	625-86-5	C_6_H_8_O	96.1	696.6	4.56	
38	2-Ethylfuran-D	3208-16-0	C_6_H_8_O	96.1	705.3	1.09	
39	2-Pentylfuran	3777-69-3	C_9_H_14_O	138.2	995.5	0.25	
Esters							
40	Ethyl Butyrate	105-54-4	C_6_H_12_O_2_	116.2	789.3	1.62	3.92
41	Ethyl 3-Hydroxybutanoate	5405-41-4	C_6_H_12_O_3_	132.2	911.7	0.67	
42	Acetic Acid Hexyl ester	142-92-7	C_8_H_16_O_2_	144.2	1010.9	0.60	
43	Methyl 2-Furoate	611-13-2	C_6_H_6_O_3_	126.1	975.8	0.51	
44	(Z)-3-Hexenyl Acetate	3681/7/18	C_8_H_14_O_2_	142.2	1011.1	0.24	
45	Methyl Salicylate	119-36-8	C_8_H_8_O_3_	152.1	1182.6	0.21	
46	Hexyl Butanoate	2639-63-6	C_10_H_20_O_2_	172.3	1198.2	0.06	
Terpenes							
47	2-Heptanone-D	110-43-0	C_7_H_14_O	114.2	890.6	0.66	3.62
48	Camphene	79-92-5	C_10_H_16_	136.2	948.7	0.48	
49	Limonene-M	138-86-3	C_10_H_16_	136.2	1038.6	0.41	
50	β-Myrcene-D	123-35-3	C_10_H_16_	136.2	993.5	0.40	
51	β-Pinene-M	127-91-3	C_10_H_16_	136.2	976.9	0.29	
52	β-Ocimene	13877-91-3	C_10_H_16_	136.2	1050.6	0.25	
53	γ-Terpinene	99-85-4	C_10_H_16_	136.2	1059.1	0.24	
54	α-Pinene-M	80-56-8	C_10_H_16_	136.2	934.1	0.23	
55	β-Myrcene-M	123-35-3	C_10_H_16_	136.2	993.3	0.16	
56	β-Myrcene-T	123-35-3	C_10_H_16_	136.2	993.1	0.14	
57	Limonene-D	138-86-3	C_10_H_16_	136.2	1038.4	0.10	
58	β-Pinene-D	127-91-3	C_10_H_16_	136.2	975.8	0.07	
59	α-Pinene-D	80-56-8	C_10_H_16_	136.2	935.1	0.07	
60	2-Heptanone-M	110-43-0	C_7_H_14_O	114.2	892	0.07	
61	β-Pinene-T	127-91-3	C_10_H_16_	136.2	976.2	0.05	
Ketones							
62	2,3-Pentadione	600-14-6	C_5_H_8_O_2_	100.1	650.4	1.93	2.35
63	2-Hydroxy-3-Methyl-2-Cyclopenten-1-One	80-71-7	C_6_H_8_O_2_	112.1	1000.6	0.28	
64	2-Pentanone	107-87-9	C_5_H_10_O	86.1	690.3	0.14	
Pyrazine							
65	2,3-Dimethylpyrazine	5910-89-4	C_6_H_8_N_2_	108.1	922.8	1.80	1.80

Note: MW, molecular weight; RI, retention index; RC, relative composition; CRC, cumulative relative content.

**Table 4 foods-13-00480-t004:** The relative contents of the 23 VOCs (VIP > 10).

No.	Compound	VIP Value	The Relative Content (%)			
			Annual *P. edulis*	Biennial *P. edulis*	Annual *C. delicatus*	Biennial *C. delicatus*
Aldehydes						
1	2-Methylpropanal	1.60	2.95 ± 0.12 ^b^	3.59 ± 0.19 ^a^	2.28 ± 0.13 ^c^	3.57 ± 0.08 ^a^
2	(E)-2-Pentenal-M	1.53	6.42 ± 0.13 ^a^	5.13 ± 0.11 ^c^	6.36 ± 0.11 ^a^	5.66 ± 0.04 ^b^
3	3-Methylbutanal-D	1.42	2.77 ± 0.14 ^c^	4.64 ± 0.18 ^a^	3.74 ± 0.04 ^b^	4.64 ± 0.03 ^a^
4	Butanal-D	1.24	1.82 ± 0.10 ^b^	1.80 ± 0.07 ^b^	1.92 ± 0.02 ^b^	2.41 ± 0.02 ^a^
5	2-Methylbutanal	1.24	3.15 ± 0.13 ^b^	4.09 ± 0.12 ^a^	3.29 ± 0.05 ^b^	4.12 ± 0.02 ^a^
6	3-Methylbutanal-M	1.14	2.39 ± 0.11 ^a^	1.28 ± 0.04 ^c^	1.51 ± 0.09 ^b^	1.40 ± 0.02 ^bc^
7	2-Phenyl-1,3-Dioxolane-4-Methanol	1.11	0.61 ± 0.02 ^b^	0.47 ± 0.03 ^c^	0.32 ± 0.02 ^d^	0.82 ± 0.01 ^a^
8	(E, E)-2,4-Heptadienal	1.08	1.05 ± 0.10 ^a^	0.07 ± 0.00 ^b^	0.09 ± 0.03 ^b^	0.07 ± 0.01 ^b^
9	Nonanal-M	1.07	0.87 ± 0.04 ^b^	0.70 ± 0.05 ^c^	0.49 ± 0.05 ^d^	0.95 ± 0.00 ^a^
10	Propanal-M	1.07	1.78 ± 0.08 ^a^	1.34 ± 0.06 ^b^	1.81 ± 0.15 ^a^	0.99 ± 0.00 ^c^
11	2,4-Hexadienal, (E, E)-D	1.03	2.31 ± 0.18 ^a^	1.66 ± 0.35 ^b^	2.23 ± 0.23 ^a^	1.94 ± 0.01 ^ab^
12	Propanal-D	1.01	0.74 ± 0.04 ^c^	1.31 ± 0.03 ^a^	0.88 ± 0.05 ^b^	0.88 ± 0.05 ^b^
Alcohols						
13	1-Propanol-M	1.63	5.18 ± 0.22 ^a^	3.05 ± 0.10 ^b^	2.99 ± 0.13 ^b^	2.34 ± 0.02 ^c^
14	2-Butanol-D	1.46	1.57 ± 0.02 ^d^	2.39 ± 0.03 ^b^	2.65 ± 0.05 ^a^	1.81 ± 0.01 ^c^
15	1-Propanethiol-D	1.07	1.96 ± 0.01 ^a^	1.42 ± 0.02 ^b^	1.08 ± 0.11 ^d^	1.30 ± 0.01 ^c^
Furans						
16	2,5-Dimethylfuran	2.42	5.34 ± 0.10 ^b^	5.00 ± 0.03 ^c^	5.54 ± 0.10 ^a^	2.42 ± 0.07 ^d^
17	2-Ethylfuran-M	2.14	1.83 ± 0.18 ^b^	5.92 ± 0.11 ^a^	5.92 ± 0.25 ^a^	6.19 ± 0.00 ^a^
18	2-Ethylfuran-D	1.11	0.59 ± 0.01 ^c^	1.44 ± 0.04 ^a^	0.81 ± 0.11 ^b^	1.53 ± 0.01 ^a^
Esters						
19	Ethyl Butyrate	1.67	1.04 ± 0.13 ^c^	2.57 ± 0.21 ^a^	1.49 ± 0.05 ^b^	1.41 ± 0.02 ^b^
20	Methyl 2-Furoate	1.11	0.27 ± 0.03 ^b^	0.23 ± 0.03 ^b^	0.77 ± 0.02 ^a^	0.78 ± 0.01 ^a^
21	Ethyl 3-Hydroxybutanoate	1.01	0.50 ± 0.05 ^c^	0.97 ± 0.03 ^a^	0.35 ± 0.07 ^d^	0.85 ± 0.02 ^b^
Terpene						
22	2-Heptanone-D	1.59	0.15 ± 0.02 ^c^	0.18 ± 0.01 ^c^	1.53 ± 0.26 ^a^	0.76 ± 0.00 ^b^
Pyrazine						
23	2,3-Dimethylpyrazine	1.70	0.88 ± 0.06 ^c^	1.28 ± 0.10 ^b^	2.53 ± 0.16 ^a^	2.49 ± 0.02 ^a^

Data are expressed as means ± SD. The small letter superscripted in each column indicates a significant difference (*p* < 0.05, *n* = 3).

## Data Availability

The original contributions presented in the study are included in the article, further inquiries can be directed to the corresponding authors.

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
