# Peer review of "Effects of the Species and Growth Stage on the Antioxidant and Antifungal Capacities, Polyphenol Contents, and Volatile Profiles of Bamboo Leaves"

_foods, 2024, doi:10.3390/foods13030480_

Round 1

Reviewer 1 Report

Comments and Suggestions for Authors

The manuscript needs some revisions/clarifications.

1. The results for the positive control in the antioxidant assays must be included in the tables.

2. A positive control is missing in the antifungal assays.

3. Phenolics and flavonoids are well-known antioxidant compounds. How the compounds identified as VOCs can act as antioxidants requires a better explanation, as normally, antioxidant molecules act as free radicals scavengers.

Comments on the Quality of English Language

The English language requires revision. Check the manuscript for misspelled words and punctuation errors.

Author Response

1. The results for the positive control in the antioxidant assays must be included in the tables.

Thank you for your kind suggestion. As mentioned in L233 and shown in Table 2, L-ascorbic acid was used as a positive control in both DPPH and ABTS+ scavenging capacity assay in this study.

Table 2. The in vitro antioxidant activities of BLE.

Sample

IC50 Value for DPPH radical (mg·mL1)

IC50 Value for ABTS+  (mg·mL1)

Annual P. edulis

0.81±0.00c

0.40±0.00c

Biennial P. edulis

0.63±0.00d

0.32±0.00d

Annual C. delicates

0.89±0.00a

0.57±0.00a

Biennial C. delicates

0.84±0.00b

0.50±0.00b

L-ascorbic acid

0.03±0.00e

0.02±0.00e

Data are expressed as mean ± SD. The small letter superscripted in each column indicates significant difference (p < 0.05, n = 3).

2. A positive control is missing in the antifungal assays.

Thank you for your excellent suggestion. We ignored to set a positive control from designing the experiment. We will take it seriously in our future research. However, we discussed the results of antifungal assays between this study and related studies.

Line 286-289: Our findings support those of Liao et al. [15], who reported that P. edulis leaf extract showed strong antifungal activity against P. capsica (inhibitory rate: 100.00%) F. graminearum (75.12%), V. mali (60.66%), B. dothidea (57.24%), V. nashicola (44.62%), and B. cinerea (30.16%).

3. Phenolics and flavonoids are well-known antioxidant compounds. How the compounds identified as VOCs can act as antioxidants requires a better explanation, as normally, antioxidant molecules act as free radicals scavengers.

Thank you for your suggestion. We also realized the contention as your comment when we revised our manuscript. In the revised manuscript, the result on Person’s correlation was mainly in the relationship between TPC, TFC and antioxidant capacity, antifungal capacity (Figure 3).

Line 24-26: Pearson’s correlation coefficients showed that the TPC was significantly negatively correlated with the half-maximal inhibitory concentrations against DPPH and ABTS+, whereas the TFC was positively correlated with C. gloeosporioides and B. cinerea growth inhibition.

Line 293-303: Antioxidant and antifungal capacities are impacted by the abundance of phenolic acids and flavonoids [5, 6, 27, 28]. Hence, we performed Pearson’s correlation analysis to identify relationships between biological activities (i.e., antioxidant and antifungal capacitiesactivities) and the abundance of biological compounds (i.e., TPC and TFC) of bamboo leaf extracts (Fig. 3). We observed significant negative correlations (p < 0.01) between the IC50 of TPC against DPPH (correlation coefficient: –1.00) and ABTS+ (–0.94). These results suggest that the TPC is a major contributor to antioxidant capacity in bamboo leaves. In addition, the TFC was highly positively correlated (p < 0.01) with C. gloeosporioides (correlation coefficient: 0.90) and B. cinerea (0.81) growth inhibition. Meanwhile, A. alternata growth inhibition was weakly correlated with both the TFC (correlation coeffi-cient: –0.23) and TPC (0.51).

Line 455-460: We compared the TPC, TFC, antioxidant and antifungal capacities of the BLEs of two bamboo species (P. edulis and C. delicatus) at two growth stages (first- and second-year leaves). Second-year P. edulis had the highest TPC, whereas second-year C. delicatus had the highest TFC. P. edulis exhibited higher antioxidant capacities, whereas C. delicatus possessed stronger antifungal capacities. Moreover, second-year bamboo leaves showed stronger biological activities than first-year leaves.

Comments on the Quality of English Language

The English language requires revision. Check the manuscript for misspelled words and punctuation errors.

The English language has been checked by at least two professional editors, both native speakers of English. For a certificate, please see:

http://www.textcheck.com/certificate/Uh0BiG

Reviewer 2 Report

Comments and Suggestions for Authors

Dear authors,

The manuscript brings the characterization of underused bamboo leaf at two different growth stages. This is useful to select the varieties for further processing.

Please revise the manuscript considering the following comments:

Introduction

L47 / L83-85 - Can the VOC be considered in the ''volatile liquids'' mentioned in L-47-48?  Specify in introduction which classes of volatiles are beneficial and which classes are harmful. I might be wrong, but VOC are generated after burning, which was not mentioned in the material studied.

MM 

L119 - Include the location with coordinates.

Topics 2.4 and 2.5 - Specify the acronyms: total phenolic content (TPC) and total flavonoid content (TFC)

l130-131 - Simplify and specify in the writing. For instance: bamboo leaf was extracted with aqueous ethanol (60:40, ethanol:water, v/v) at the leaf:solvent proportion of 1:20 (v/v). Make sure this type of correction is done througout the text

Topic 3.1 - The authors measured antioxidant capacity in vitro, not the activity. Correct the names throughout the text.

RD section

Organize the ideas at same order as MM section, i.e., explain total phenolic content and total flavonoid content first, followed by antioxidant capacity

L 333-335 - Which molecule may induce higher antioxidant capacity or which molecule may induce oxidation? Cite more examples as you did in L-357.

Additional comments:

Specify which type of plant growth stage was in the first and second year (vegetative? flowering?...)

Highlights the novelties provided by this work and how the findings would be useful to select the best variety in terms of bioactive compounds.

Author Response

The manuscript brings the characterization of underused bamboo leaf at two different growth stages. This is useful to select the varieties for further processing.

Please revise the manuscript considering the following comments:

  1. Introduction

L47 / L83-85 - Can the VOC be considered in the “volatile liquids” mentioned in L-47-48? Specify in introduction which classes of volatiles are beneficial and which classes are harmful. I might be wrong, but VOC are generated after burning, which was not mentioned in the material studied.

Thank you for your kind suggestion.

(1) Can the VOC be considered in the “volatile liquids” mentioned in L-47-48?

We have substitude “phenolic and flavonoid compounds” for “volatile liquids”.

Line 46-48: The inner cavity of most species in the Chimonocalamus subfamily contains phenolic and flavonoid compounds, which confer its biologically active properties.

(2) Specify in introduction which classes of volatiles are beneficial and which classes are harmful.

To our best knowledge, at present, there is not enough evidence to prove which volatiles in bamboo leaves are harmful or beneficial.

 (3)  I might be wrong, but VOC are generated after burning, which was not mentioned in the material studied.

Line 181-187: 2.8. GC-IMS assay

The volatile profiles of the four types of bamboo leaves were analyzed using a FlavourSpec® GC-IMS system (G.A.S Company, Berlin, Germany) as described by Shen et al. [20], with adjustments. Fresh bamboo leaves were ground, and a 2 g aliquot was transferred to a headspace bottle (20 mL) and incubated at 60°C for 20 min. Next, 200 µL sample solution was automatically injected using a heated syringe needle (85°C) before incubating at 60°C for 20 min with 500 r min–1.

  1. MM

L119 - Include the location with coordinates.

Thank you for your kind suggestion. We supplied the detail information of the sample location.

Line 118-122: The annual and biennial leaves of P. edulis and C. delicatus were harvested from Zhenzhu Agricultural Co. Ltd. (Upper Left Longitude: 103.088665°, Upper Left Latitude: 25.337019°; Lower Left Longitude: 103.089867°, Lower Left Latitude: 25.333840°; Upper Right Longitude: 103.092313°, Upper Right Latitude: 25.336980°; Lower Right Longitude: 103.091948°, Lower Right Latitude: 25.334829°) in April 2022 as the experimental materials (Fig. 1).

Topics 2.4 and 2.5 - Specify the acronyms: total phenolic content (TPC) and total flavonoid content (TFC)

We have specified the acronyms.

Line 136: 2.4. Total phenolic content (TPC) determination

Line 145: 2.5. Total flavonoid content (TFC) determination

l30-131 - Simplify and specify in the writing. For instance: bamboo leaf was extracted with aqueous ethanol (60:40, ethanol:water, v/v) at the leaf:solvent proportion of 1:20 (v/v). Make sure this type of correction is done througout the text.

Thank you for your excellent suggestion. We have corrected the type throughout the text.

Line 133-134: Samples of dried bamboo leaf powder were extracted twice with aqueous ethanol (60:40, ethanol:water, v/v).

Line 139: Bamboo leaf was extracted twice with aqueous ethanol (60:40, ethanol:water, v/v) at the leaf:solvent proportion of 1:20 (v/v).

Line 141-142: Next, a mixture of 1.5 mL aqueous sodium carbonate (20:80, sodium carbonate:water, w/v) and 0.8 mL water was added and incubated for 90 min.

Topic 3.1 - The authors measured antioxidant capacity in vitro, not the activity. Correct the names throughout the text.

Thank you for your suggestion. We have corrected the names throughout the text.

  1. RD section

Organize the ideas at same order as MM section, i.e., explain total phenolic content and total flavonoid content first, followed by antioxidant capacity.

We have organized the ideas at same order as MM section.

3.1. TPC and TFC of BLE

3.2. Antioxidant capacities against DPPH and ABTS+ radicals

3.3. Antifungal capacities against C. gloeosporioides, B. cinerea, and A. alternata

L333-335 - Which molecule may induce higher antioxidant capacity or which molecule may induce oxidation? Cite more examples as you did in L-357.

Thank you for your kind suggestion.

In this study, the result on Person’s correlation was mainly in the relationship between TPC, TFC and antioxidant capacity, antifungal capacity. The exact relationship between VOCs and antioxidant and antifungal capacities will be assessed in the further research.

Additional comments:

Specify which type of plant growth stage was in the first and second year (vegetative? flowering?...)

We have specified the geographical coordinates and harvest time of the bamboo leaves.

Line 37: Bamboos are perennial, evergreen plant species in the Gramineae family. Therefore, they don’t have obvious plant growth stage such as vegetative, flowering, ...

Line 118-122: The annual and biennial leaves of P. edulis and C. delicatus were harvested from Zhenzhu Agricultural Co. Ltd. (Upper Left Longitude: 103.088665°, Upper Left Latitude: 25.337019°; Lower Left Longitude: 103.089867°, Lower Left Latitude: 25.333840°; Upper Right Longitude: 103.092313°, Upper Right Latitude: 25.336980°; Lower Right Longitude: 103.091948°, Lower Right Latitude: 25.334829°) in April 2022 as the experimental materials (Fig. 1).

Highlights the novelties provided by this work and how the findings would be useful to select the best variety in terms of bioactive compounds.

Thank you for your comment. The Highlights of this work were as follows:

1) Antioxidant and antifungal capacity of BLE varied with genotype and growth period;

2) TPC and TFC varied with genotype and growth period;

3) TPC was negative correlated with antioxidant capacities;

4) TFC was positive correlation with antifungal capacities;

5) Discriminative VOCs with VIP higher than 1 were screened.

Our results provide a theoretical foundation to promote the use of bamboo leaves by discriminate bamboo leaves in different species, growth stage based on their difference in biological activities, TPC, TFC and volatile profile.
